# Comparative efficacy of terlipressin and norepinephrine for treatment of hepatorenal syndrome-acute kidney injury: A systematic review and meta-analysis

Jody C. Olson[1]*, Ram M. Subramanian[2]

1 Division of Gastroenterology and Hepatology, Mayo Clinic, Rochester, Minnesota, United States of America, 2 Department of Medicine, Divisions of Gastroenterology and Hepatology and Pulmonary and Critical Care Medicine, Emory University School of Medicine, Atlanta, Georgia, United States of America

* olson.jody3@mayo.edu

**Data Availability Statement:** All data generated or analyzed in this meta-analysis are included within

## Abstract

The treatment of choice for hepatorenal syndrome-acute kidney injury (HRS-AKI) is vasoconstrictor therapy in combination with albumin, preferably norepinephrine or terlipressin as recommended by recent guidelines. In the absence of larger head-to-head trials comparing the efficacy of terlipressin and norepinephrine, meta-analysis of smaller studies can provide insights needed to understand the comparative effects of these medications. Additionally, recent changes in the HRS diagnosis and treatment guidelines underscore the need for newer analyses comparing terlipressin and norepinephrine. In this systematic review, we aimed to assess reversal of hepatorenal syndrome (HRS) and 1-month mortality in subjects receiving terlipressin or norepinephrine for the management of HRS-AKI. We searched literature databases, including PubMed, Cochrane, Clinicaltrials.gov, International Clinical Trials Registry Platform, Embase, and ResearchGate, for randomized controlled trials (RCTs) published from January 2007 to June 2023 on June 26, 2023. Only trials comparing norepinephrine and albumin with terlipressin and albumin for the treatment of HRS-AKI in adults were included, and trials without HRS reversal as an endpoint or nonresponders were excluded. Pairwise meta-analyses with the random effects model were conducted to estimate odds ratios (ORs) for HRS reversal and 1-month mortality as primary outcomes. Additional outcomes assessed included HRS recurrence, predictors of response, and incidence of adverse events (AEs). We used the Cochrane risk of bias assessment tool for quality assessment. We included 7 RCTs with a total of 376 subjects with HRS-AKI or HRS type 1. This meta-analysis showed numerically higher rates of HRS reversal (OR 1.33, 95% confidence interval [CI] [0.80–2.22]; $P = 0.22$) and short-term survival (OR 1.50, 95% CI [0.64–3.53]; $P = 0.26$) with terlipressin, though these results did not reach statistical significance. Terlipressin was associated with AEs such as abdominal pain and diarrhea, whereas norepinephrine was associated with cardiovascular AEs such as chest pain and ischemia. Most of the AEs were reversible with a reduction in dose or discontinuation of therapy across both arms. Of the terlipressin-treated subjects, 5.3% discontinued therapy due to serious AEs compared to 2.7% of the norepinephrine-treated

this article and cited in the references section. The included references are publicly available.

**Funding:** This analysis as well as editorial and writing support were funded by Mallinckrodt Pharmaceuticals, plc, Bridgewater, NJ, US. The funders had no role in study design, data collection and analysis, decision to publish, or preparation of the manuscript.

**Competing interests:** Both JCO and RMS have served as consultants for Mallinckrodt Pharmaceuticals related to terlipressin. This does not alter our adherence to PLOS ONE policies on sharing data and materials. There are no patents, products in development or marketed products associated with this research to declare.

**Abbreviations:** AASLD, American Association for the Study of Liver Diseases; ACLF, acute-on-chronic liver failure; AE, adverse event; AKI, acute kidney injury; CI, confidence interval; EASL, European Association for the Study of the Liver; HRS, hepatorenal syndrome; HRS-1, hepatorenal syndrome type 1; HRS-2, hepatorenal syndrome type 2; HRS-AKI, hepatorenal syndrome-acute kidney injury; HRS-CKD, hepatorenal syndrome-chronic kidney disease; ICA, International Club of Ascites; ICU, intensive care unit; IV, intravenous; MAP, mean arterial pressure; MELD, model for end-stage liver disease; NR, not reported; OR, odds ratio; RCT, randomized controlled trial; SCr, serum creatinine; SE, standard error.

subjects. Limitations of this analysis included small sample size and study differences in HRS-AKI diagnostic criteria. As more studies using the new HRS-AKI criteria comparing terlipressin and norepinephrine are completed, a clearer understanding of the comparability of these 2 therapies will emerge.

## Introduction

Hepatorenal syndrome (HRS) is traditionally defined as renal failure resulting from vasoconstriction and hemodynamic changes occurring in cirrhotic patients with ascites and portal hypertension [1]. The diagnosis and staging of HRS has changed in recent guidelines, the new HRS classification includes 2 types, acute kidney injury (HRS-AKI) and chronic kidney injury (HRS-CKD) [1,2]. HRS-AKI, previously known as HRS type 1 (HRS-1), occurs in patients with cirrhosis in the absence of hypovolemia or significant abnormalities in kidney histology [1]. The primary difference between the definitions of HRS-1 and HRS-AKI is the elimination of an absolute serum creatinine (SCr) threshold in the latter [1,3]. The current definition of HRS-CKD, on the other hand, now encompasses patients previously diagnosed as HRS type 2 (HRS-2) [1,2].

The prevalence of HRS-AKI in hospitalized patients with decompensated cirrhosis ranges between 27% and 53%, and the development of acute kidney injury (AKI) is associated with a poor prognosis and high 30-day mortality ranging from 29% to 44% [1]. Vasoconstrictor drugs in combination with albumin are the treatment of choice for HRS-AKI to counteract the splanchnic arterial vasodilation and improve effective circulating volume with resultant improvement in renal perfusion [1,2].

Several studies have shown that vasoconstrictors, specifically terlipressin or norepinephrine, in combination with albumin are effective in improving kidney function in patients with HRS-AKI [1,2,4–7]. However, large head-to-head trials comparing the efficacy of norepinephrine and terlipressin have not been conducted, making it difficult to evaluate the comparative efficacy of the 2 vasoconstrictors. Both the American Association for the Study of Liver Diseases (AASLD) Guidance and the European Association for the Study of the Liver (EASL) guidelines list terlipressin as the preferred treatment for HRS-AKI and recommend norepinephrine as an alternative, while the American College of Gastroenterology guidelines recommend either terlipressin or norepinephrine as first-line treatment options [1,2,4]. Given the recent approval of terlipressin in the US and the historical use of norepinephrine in patients with HRS-AKI, there is a need among US physicians to evaluate the comparative efficacy of these treatment options commonly used as first-line therapy [8]. Since there are limited studies with direct comparison of terlipressin and norepinephrine, a meta-analysis may provide healthcare practitioners with relevant data to guide HRS-AKI patient management.

Three meta-analyses evaluating the efficacy of multiple vasoactive therapies for HRS-1 or HRS-2 have been previously published [9–11]. Data from Facciorusso et al suggested that the use of terlipressin or norepinephrine was more efficacious when compared to midodrine plus octreotide for HRS reversal [9]. They also found that terlipressin may reduce short-term mortality compared to placebo; but neither terlipressin nor norepinephrine was associated with a significant mortality benefit over any comparator [9]. Zheng et al found terlipressin to be the most efficacious vasoconstrictor for reversal of HRS and short-term mortality when compared to norepinephrine and midodrine plus octreotide [11]. However, terlipressin was also associated with increased risk of adverse events (AEs) [11]. Meanwhile, norepinephrine appeared to

be an appropriate alternative with lower risk of AEs [11]. Furthermore, Wang et al found terlipressin and norepinephrine to be comparable for HRS reversal [10].

Since these meta-analyses, additional studies on the comparative efficacy of the 2 treatments have been published [9–14]. Thus, with the addition of this new evidence, we conducted a meta-analysis to compare the reported efficacy of terlipressin and norepinephrine for the treatment of HRS-1 or HRS-AKI.

## Methods

### Search strategy and selection criteria

We conducted literature searches of PubMed, Cochrane, Clinicaltrials.gov, International Clinical Trials Registry Platform, Embase, and ResearchGate databases for randomized controlled trials (RCTs) published between January 2007 and June 2023 on June 26, 2023. The search strategy included the Boolean terms such as "hepatorenal syndrome" or "hepatorenal syndrome-acute kidney injury," and "norepinephrine" or "noradrenaline," and "terlipressin." We included for evaluation all RCTs with adults aged ≥18 years who were diagnosed with HRS-1 or HRS-AKI and focused specifically on trials that conducted a direct comparison of norepinephrine or terlipressin in combination with albumin, since these 2 vasoconstrictors are most recommended by the guidelines.

The major outcomes assessed were HRS reversal and short-term mortality for patients with HRS-AKI. Additional outcomes assessed were HRS recurrence, predictors of response, and safety. We excluded trials in which HRS reversal was not an endpoint as well as trials in which subjects who did not respond to initial vasoconstrictor therapy were included, since this may skew the results. We also excluded meta-analysis or systematic reviews, observational studies, retrospective studies, trials with subjects who had dual diagnoses such as sepsis or cardiopulmonary diseases in addition to HRS as well as any study published in languages other than English. Two independent reviewers carried out our search strategy. We manually evaluated the titles and abstracts of trials to exclude those that did not meet the inclusion criteria. Two independent reviewers evaluated the eligibility of the remaining trials. Please see the Supplemental Information for additional details regarding the selection of the studies and the reason for inclusion or exclusion in this review.

### Data extraction and outcomes

Three reviewers extracted the data independently from each study. We evaluated outcomes related to HRS reversal and short-term mortality. The primary outcome in each trial was reversal of HRS as defined by complete response. Five of the 7 trials defined complete response as a decrease in SCr to a value of 1.5 mg/dL or lower during treatment [14–18]. One trial, Indrabi et al, did not define complete response in the abstract [19]. The Arora et al trial defined complete response as return of SCr to a value within 0.3 mg/dL of baseline as reflected in the new AASLD Guidance [1,12].

Reported short-term mortality ranged from 14 days to 90 days in all trials. We opted to assess 1-month mortality, defined as either 28-day or 30-day mortality. We excluded trials, such as Indrabi and Goyal et al, from our assessment since they did not evaluate mortality at 1 month [16,19]. We also evaluated other outcomes in this meta-analysis that were reported as secondary outcomes in the trials, such as recurrence of HRS after initial reversal, predictors of response, and incidence of AEs. When trials included subjects with HRS-1 and HRS-2, we selectively extracted data for subjects with HRS-1 where reported.

## Quality assessment

To assess the risk of bias, 2 reviewers independently evaluated the studies using the Cochrane risk of bias assessment tool. This tool evaluated 5 domains of bias for randomized trials: bias arising from the randomization process, bias due to deviations from intended interventions, bias due to missing outcome data, bias in measurement of the outcome, and bias in selection of the reported result.

## Data analysis

We performed pairwise meta-analysis with a random effects model to calculate 95% confidence intervals (CIs) and pooled estimates of odds ratios (ORs) to evaluate HRS reversal and 1-month mortality outcomes. The RCTs used to conduct this analysis had variation in treatment intensity, length of treatment, and target population. Therefore, due to the anticipated heterogeneity, we used a random effects model with Knapp-Hartung adjustment to evaluate the pooled effect sizes between the trials and their respective standard errors. We also assessed between-study heterogeneity using the $I^2$ and $\tau^2$ statistics to understand how the true effect sizes vary within our meta-analysis study. Additionally, we decided to use the frequentist inference approach since it is more commonly used than the Bayesian counterpart and has better interpretability. The analysis was done with R version 4.2.2.

## Results

### Study characteristics

The database search identified 189 records, 25 of which were duplicates. This resulted in 164 unique records that were individually screened for eligibility. Seven RCTs, with a total of 376 participants diagnosed with HRS-1 or HRS-AKI, comparing terlipressin and norepinephrine, were included in the meta-analysis review (Fig 1).

A few trials that initially met our inclusion criteria were excluded upon further evaluation. Of note, a recent trial by Nayyar et al met the inclusion criteria for this meta-analysis but was excluded because results fell significantly outside of the normal distribution when plotted on the forest plot matrix [13]. In addition, the 0% HRS reversal rate in the norepinephrine arm prevented calculation of an effect size unless corrections were used [13]. Other studies such as Jha et al and Ullah et al evaluating terlipressin and norepinephrine were not included since they did not evaluate HRS reversal as an endpoint [20,21]. In addition, Koneti et al was not included since participants received midodrine along with terlipressin or norepinephrine in the trial [22].

All trials included in this study were open-label RCTs, except Indrabi et al, which provided no information regarding its blinding protocol [12,14–19]. All RCTs were conducted exclusively in subjects with HRS-1 or provided data separately for a subset of HRS-1 subjects if the population included subjects diagnosed with either HRS-1 or HRS-2. The duration of treatment ranged from 14 days to 15 days in all trials [12,14–18], except for Indrabi et al [19], which did not disclose the treatment duration in its abstract. All trials reported HRS reversal rate and mortality; however, other secondary endpoints such as recurrence of HRS, predictors of response, and incidence of AEs were reported inconsistently. All participants received supportive therapy with albumin. Additionally, 2 trials, Sharma et al and Goyal et al, administered third-generation cephalosporin prophylactically as supportive therapy [16,17]. It was also noted that Goyal et al administered furosemide to maintain urine output in norepinephrine-treated subjects only [16]. Terlipressin was administered as intravenous (IV) bolus in 5 trials [14–18], and as continuous IV infusion in 1 trial [12], while Indrabi et al [19] did not specify

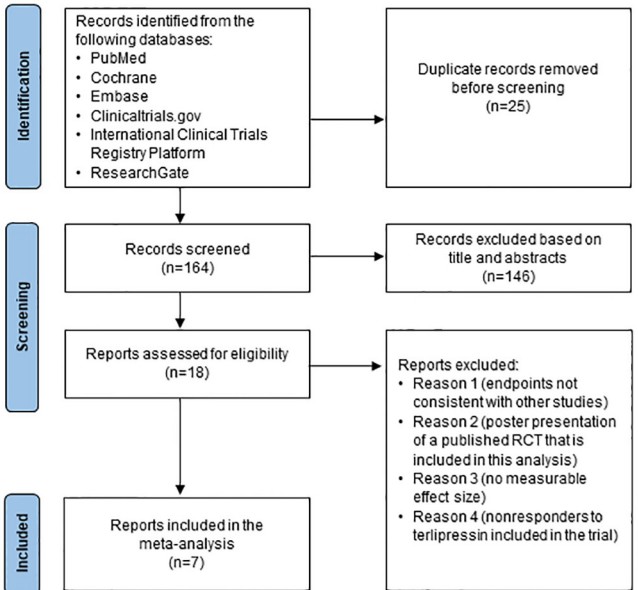

**Fig 1. Flow diagram of the study selection process.** RCT = randomized controlled trial.

its route of administration. Table 1 shows the summary of all study parameters included in this meta-analysis. Baseline characteristics were similar across trials as shown in Table 2.

## HRS reversal

The overall HRS reversal rate was higher for subjects treated with terlipressin at 47.9% (90/188) compared to those treated with norepinephrine at 39.9% (75/188). However, these results were not statistically significant (OR 1.33; 95% CI [0.80–2.22]; $P$ = 0.22) (Fig 2). There was no significant heterogeneity among studies ($I^2$ = 0%, $P$ = 0.45; Fig 2).

## Mortality

All 7 trials included in the meta-analysis reported mortality at varying times, ranging from 14 days to 90 days post-treatment initiation. Five of the 7 trials reported 28-day or 30-day mortality and were included in the 1-month mortality assessment (275 subjects) [12,14,15,17,18]. Goyal et al and Indrabi et al trials were not included in the 1-month mortality outcome since they reported mortality at 14 days and 90 days, respectively [16,19]. The 1-month mortality rate was lower for those receiving terlipressin (50.7% [70/138]) compared to those receiving norepinephrine (63.5% [87/137]) across the 5 trials included [12,14,15,17,18]. However, the result was not statistically significant (OR 1.50, 95% CI [0.64–3.53]; $P$ = 0.26; Fig 3). Moderate heterogeneity, although insignificant, was noted among the included studies (Heterogeneity $I^2$ = 36%, $P$ = 0.18; Fig 3) [24].

## Additional endpoints

Meta-analysis of additional endpoints could not be performed as these were inconsistently reported across trials. The most common secondary endpoints included HRS recurrence and predictors of response. Out of the 2 trials reporting HRS recurrence, 5 (7.5%) subjects, consisting of 2 (3.0%) subjects from norepinephrine arms and 3 (4.5%) subjects from terlipressin

**Table 1. Parameters of included studies.**

| | Study design | Location; year; follow-up length | HRS-1 or HRS-AKI diagnostic criteria | Intervention (no. of participants) | Concomitant therapies | Relevant outcomes reported | No. (%) HRS reversal | No. (%) mortality at 28 days or 30 days |
|---|---|---|---|---|---|---|---|---|
| Alessandria et al, 2007 [15] | Single-center, open-label, RCT | Italy; NR; 6 months | HRS-1 diagnostic criteria: International Club of Ascites (ICA) 2007[a] | Norepinephrine (continuous infusion) 0.1 μg/kg per min up to 0.7 μg/kg/min until HRS reversal or for a maximum of 14 days (n = 4) Terlipressin (IV bolus) 1 mg every 4 h up to 2 mg every 4 h until HRS reversal or for maximum 14 days (n = 5) | Albumin 20 g/100 mL | HRS reversal[c]; 1-month mortality; HRS recurrence; incidence of side effects | Norepinephrine 3/4 (75) Terlipressin 4/5 (80) | Norepinephrine 1/4 (25) Terlipressin 1/5 (20) |
| Sharma et al, 2008 [17] | Single-center, open-label, RCT | India; 2005–2006; 15 days | HRS-1 diagnostic criteria: ICA 2007[a] HRS-1: Defined as SCr level >2.5 mg/dL or a 24-hour creatinine clearance level <20 mL/min | Norepinephrine (continuous infusion) 0.5 mg/h up to 3 mg/h for 15 days (n = 20) Terlipressin (IV bolus) 0.5 mg every 6 h up to 2 mg every 6 h for 15 days (n = 20) | Albumin 20–40 g/day Third-generation cephalosporins prophylactically | HRS reversal[d]; 1-month mortality; predictors of response; and incidence of side effects | Norepinephrine 10/20 (50) Terlipressin 10/20 (50) | Norepinephrine 9/20 (45) Terlipressin 9/20 (45) |
| Singh et al, 2012 [18] | Single-center, open-label, RCT | India; 2009–2011; 30 days | HRS-1 diagnostic criteria: Cirrhosis with ascites with SCr levels ≥2.5 mg/dL; absence of shock; fluid losses and treatment with nephrotoxic drugs; no improvement in renal function following diuretic withdrawal and plasma volume expansion; no ultrasound evidence of renal parenchymal disease or obstructive uropathy, and absence of proteinuria more than 500 mg/24 h | Norepinephrine (continuous infusion) 0.5 mg/h up to 3 mg/h until HRS reversal or a maximum of 15 days (n = 23) Terlipressin (IV bolus) 0.5 mg every 6 h up to 2 mg every 6 h until HRS reversal or for a maximum of 15 days (n = 23) | Albumin 20 g/day | HRS reversal[e]; 1-month mortality; predictors of response, and incidence of side effects | Norepinephrine 10/23 (43.4) Terlipressin 9/23 (39.1) | Norepinephrine 15/23 (65.2) Terlipressin 16/23 (69.5) |
| Indrabi et al, 2013 [19] | Single-center, RCT | India; NR; NR | No information provided | Norepinephrine, dose, route of administration, and duration NR (n = 30) Terlipressin, dose, route of administration, and duration NR (n = 30) | Albumin (dose NR) | HRS reversal[f]; predictors of response, and HRS recurrence | Norepinephrine 16/30 (53) Terlipressin 17/30 (57) | NR |

(*Continued*)

**Table 1.** (Continued)

| | Study design | Location; year; follow-up length | HRS-1 or HRS-AKI diagnostic criteria | Intervention (no. of participants) | Concomitant therapies | Relevant outcomes reported | No. (%) HRS reversal | No. (%) mortality at 28 days or 30 days |
|---|---|---|---|---|---|---|---|---|
| Goyal et al, 2016 [16] | Single-center, open-label, RCT | India; NR; NR | HRS-1 diagnostic criteria: ICA 2007[a] HRS-1: Defined as doubling of initial SCr to >2.5 mg/dL in less than 2 weeks | Norepinephrine (continuous infusion) 0.5 mg/h up to 3 mg/h until HRS reversal or maximum 14 days (n = 21) Terlipressin (IV bolus) 0.5 mg every 6 h up to 2 mg every 6 h until HRS reversal or maximum 14 days (n = 20) | Albumin 20 g/day Furosemide 0.001 mg/kg/min if adequate urine output was not achieved (norepinephrine arm only) Third-generation cephalosporin prophylactically | HRS reversal[d]; predictors of response; and incidence of side effects | Norepinephrine 10/21 (47.6) Terlipressin 9/20 (45) | NR |
| Saif et al, 2018 [14] | Single-center, open-label, RCT | India; NR; 90 days | HRS-1 diagnostic criteria: ICA 2007[a] | Norepinephrine (continuous infusion) 0.5 mg/h up to 3 mg/h until HRS reversal or 14 days of therapy (n = 30) Terlipressin (IV bolus) 0.5 mg every 6 h up to 2 mg every 6 h until HRS reversal or 14 days of therapy (n = 30) | Albumin 20–40 g/day | HRS reversal[d]; 1-month mortality; predictors of response | Norepinephrine 16/30 (53) Terlipressin 17/30 (57) | Norepinephrine 14/30 (46.7) Terlipressin 13/30 (43.3) |
| Arora et al, 2020 [12] | Single-center, open-label, RCT | India; Oct 2015-Dec 2016; NR | HRS-AKI diagnostic criteria: ICA 2015[b] | Norepinephrine (continuous infusion) 0.5 mg/h up to 3 mg/h until HRS reversal or 14 days of therapy (n = 60) Terlipressin (continuous infusion) 2 mg/day up to 12 mg/day until HRS reversal or 14 days of therapy (n = 60) | Albumin 20–40 g/day | HRS reversal[g]; 28-day mortality; predictors of response; incidence of side effects | Norepinephrine 10/60 (16.7) Terlipressin 24/60 (40%) | Norepinephrine 48/60 (80%) Terlipressin 31/60 (51.7) |

NR = not reported.

[a] Presentation with cirrhosis and ascites, SCr >1.5 mg/dL, with no improvement of SCr after at least 2 days of diuretic withdrawal and volume expansion with albumin at the dose of 1 g/kg of body weight per day (maximum of 100 g/day), absence of shock, no current or recent treatment with nephrotoxic drugs, and absence of parenchymal kidney disease as indicated by proteinuria >500 mg/day, microhematuria (>50 red blood cells per high power field), and/or abnormal renal ultrasonography.

[b] Diagnosis of cirrhosis with ascites; diagnosis of AKI according to ICA-AKI (ICA-AKI stage ≥II); no response after 2 consecutive days of diuretic withdrawal and plasma volume expansion with albumin 1 g/kg of body weight per day; absence of shock; no current or recent use or nephrotoxic drugs; no macroscopic signs of structural kidney injury, defined as absence of proteinuria (>500 mg/day), absence of microhematuria (>50 red blood cells per high power field), normal findings on renal ultrasonography [23].

[c] Defined as decrease of SCr level ≥30% compared with the baseline value to a final value of ≤1.5 mg/dL during the treatment.

[d] Defined as decrease in SCr to a value of ≤1.5 mg/dL during the treatment.

[e] Defined as SCr <1.5 mg/dL.

[f] No information provided.

[g] Defined as return of SCr to a value within 0.3 mg/dL of baseline. Baseline value of SCr obtained in the previous 3 months.

**Table 2. Baseline characteristics of the subjects across included studies.**

| Study | MELD score | | MAP (mmHg) | | SCr (mg/dL) | |
|---|---|---|---|---|---|---|
| | Terlipressin | Norepinephrine | Terlipressin | Norepinephrine | Terlipressin | Norepinephrine |
| Alessandria et al [15] | 26 ± 2 | 26 ± 1 | 74 ± 3 | 71 ± 2 | 2.5 ± 0.3 | 2.3 ± 0.2 |
| Sharma et al [17] | 29.6 ± 6.2 | 31.6 ± 6 | 81.4 ± 11.4 | 78.2 ± 5.3 | 3.0 ± 0.5 | 3.3 ± 1.3 |
| Singh et al [18] | 26.4 | 24.7 | 64.7 | 65.2 | 3.3 | 3.1 |
| Indrabi et al [19] | No information | No information | No information | No information | No information | No information |
| Goyal et al [16] | 30.1 ± 5.9 | 29.2 ± 6.1 | 76.8 ± 11.6 | 77.3 ± 8.6 | 3.4 ± 1.6 | 3.1 ± 1.5 |
| Saif et al [14] | 29.1 ± 5.8 | 30.4 ± 9.2 | 81.3 ± 8.1 | 80.6 ± 10.2 | 3.2 ± 0.9 | 3.3 ± 1.1 |
| Arora et al [12] | 33.3 ± 5.0 | 33.8 ± 5.0 | 68.1 ± 4.6 | 67.9 ± 4.2 | 1.8 (1.09, 5.30) | 2.0 (1.02–5.10) |

MAP = mean arterial pressure, MELD = model for end-stage liver disease.

arms, experienced HRS recurrence [15,19]. Alessandria et al defined HRS recurrence as an increase in SCr level of 50% or more than the lowest value after treatment in subjects with complete response with a final value above 1.5 mg/dL during the follow-up period, while Indrabi et al did not define HRS recurrence [15,19].

Six of the 7 trials included in this meta-analysis reported predictors of response as an additional outcome [12,14,16–19]. Common predictors of response reported for both arms were Child-Turcotte-Pugh score, model for end-stage liver disease (MELD) score, serum urea, serum albumin, and prothrombin time on multivariate analysis for multiple trials [12,14,16–19].

## Adverse events

All trials inconsistently reported data for treatment-related AEs. Common AEs reported in the terlipressin arms included abdominal cramps and increased frequency of stools, which improved with a decrease in terlipressin dose. In norepinephrine arms, commonly reported AEs included cardiovascular events such as chest pain and ventricular ectopy without hemodynamic compromise. Across all studies, approximately 5.3% (10/188) of subjects treated with terlipressin required discontinuation of therapy due to serious AEs compared to 2.7% (5/188) of norepinephrine-treated subjects. Three trials reported cardiovascular events such as ST segment depression, chest pain, ventricular ectopy, and peripheral ischemia, affecting subjects in both terlipressin and norepinephrine arms [16–18].

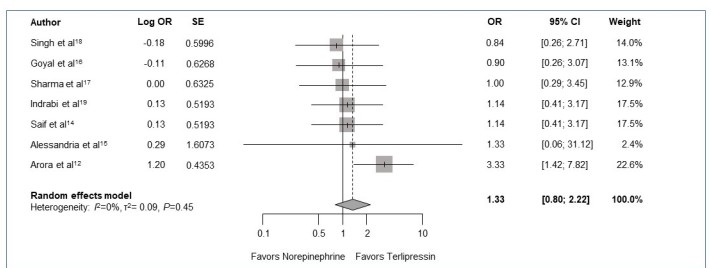

**Fig 2. Meta-analysis comparing HRS reversal of norepinephrine and terlipressin.** Forest plot matrix that compares reported HRS reversal data of all included trials using OR and heterogeneity calculated with a random effects model by pairwise meta-analysis. *P* value for overall effect *P* = 0.22. CI = confidence interval, OR = odds ratio, SE = standard error.

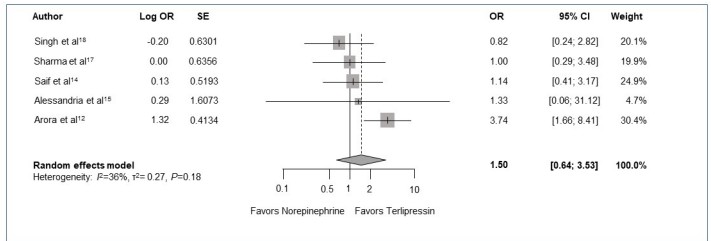

**Fig 3. Meta-analysis comparing 1-month mortality of norepinephrine and terlipressin.** Forest plot matrix that compares reported 1-month mortality data of all included trials using OR and heterogeneity calculated with a random effects model by pair-wise meta-analysis. *P* value for overall effect *P* = 0.26. CI = confidence interval; OR = odds ratio; SE = standard error.

## Quality assessment

Overall, our evaluation indicated that most of the trials included in this meta-analysis were at either "low risk," [12,17] or had "some concerns," [14–16,18] of bias using the Cochrane risk of bias assessment tool. Indrabi et al was the only trial determined to be at "high risk" of bias since only the trial abstract was available [19]. One trial [19], did not report its randomization process, and most of the trials [14–17,19], did not report their allocation concealment methods. Nevertheless, this did not result in high risk of bias since baseline characteristics were evenly distributed between the 2 treatment arms. In most of the trials, participants and healthcare team were aware of the assigned intervention during the trial. However, this did not result in high risk of bias since blinding is unlikely to impact objective outcomes measured in the trials such as HRS reversal (change in SCr) or mortality. Please see the Supplemental Information for additional details on reviewers' assessments of the risk of bias in the included studies.

## Discussion

HRS-AKI is characterized by a progressive and rapid deterioration of renal function, which often leads to multiorgan failure and death [25]. Liver transplantation is the optimal treatment for patients with HRS-AKI, even among those who respond to vasoconstrictor therapy [1,2]. The AASLD Guidance and EASL guidelines recommend terlipressin as the first-line option or, alternatively, norepinephrine in cases where terlipressin is not available [1,2]. Our data support this recommendation, with terlipressin having higher rates of both HRS reversal and short-term survival, although the observed ORs for these endpoints did not achieve statistical significance, suggesting norepinephrine is an appropriate option as noted in the guidelines. Observed AEs associated with terlipressin in the included trials were abdominal cramps and increased frequency of stools, whereas the AEs associated with norepinephrine included cardiovascular events such as chest pain and ischemia. The recent phase 3 CONFIRM trial, which evaluated the efficacy of terlipressin versus placebo for the treatment of HRS, also reported these AEs associated with terlipressin [7]. However, in the CONFIRM trial, an additional significant AE of concern was respiratory failure, which was not observed in the included trials. A variety of factors contribute to this finding, including differences in HRS-AKI diagnostic criteria that delayed the timing of terlipressin administration, the presence of acute-on-chronic liver failure (ACLF) with associated multiorgan dysfunction, and cumulative albumin dosage. Further studies will provide insights into how these factors affect the risk of respiratory failure associated with terlipressin use.

We conducted this systematic review because several new studies comparing safety and efficacy of terlipressin and norepinephrine have been published since the last meta-analysis.

Facciorusso et al and Zheng et al included comparative efficacy trials for all available vasoactive therapies among patients with HRS-1, while Wang et al compared terlipressin with all available vasoconstrictor therapies as well as placebo and albumin among patients with HRS-1 and patients with HRS-2 [9–11]. In contrast, our meta-analysis assessed only the efficacy of terlipressin and norepinephrine, as they are guideline-recommended treatments for HRS-AKI with superiority over other vasoconstrictor therapies such as midodrine and octreotide [1,2].

In our assessment of the new studies being added to this meta-analysis, we chose to remove the study by Nayyar et al because the results fell significantly outside of the normal distribution in favor of terlipressin. Even without this study, our conclusion is similar to that reported by Zheng et al, who also reported that the difference between norepinephrine and terlipressin was not statistically significant but used a graphical representation of SUCRA probability to demonstrate that terlipressin was considered the most effective for HRS reversal. The other 2 meta-analyses have found terlipressin and norepinephrine to be comparable [9–11]. The difference in our conclusions can be attributed to the inclusion of newer RCTs in our meta-analysis.

One such publication, a recent trial by Arora et al, used the updated HRS-AKI definition for diagnosis and restricted the subject population to those experiencing ACLF [12]. ACLF is characterized by acute decompensation in patients with cirrhosis, usually precipitated by an event (commonly bacterial infection) that leads to multiple organ failures, including AKI. Patients with ACLF are at high risk of short-term mortality [26,27]. ACLF criteria are increasingly being used by hepatologists and critical care specialists for management of patients with chronic liver disease in the critical care setting. In subjects with ACLF enrolled in the trial by Arora et al, terlipressin improved HRS reversal, significantly reduced the need for renal replacement therapy, and improved 28-day survival compared to norepinephrine [12]. The authors of this study also noted that while there were more AEs limiting the use of the drug in the terlipressin arm, the events were reversible. It is interesting to note that in our meta-analysis, the HRS reversal results from the Arora et al trial are the only results for which all values within the margin of error favor terlipressin, indicating that this subject population may experience a clear benefit from treatment with terlipressin over norepinephrine compared to a broader patient population [12]. Terlipressin's higher selectivity for vasopressin 1 receptors over vasopressin 2 receptors promotes vasoconstriction in both the systemic and splanchnic circulation, making it a promising therapeutic option to combat the severe inflammatory response and circulatory impairment in patients with ACLF. In addition, in the setting of both systemic and splanchnic vasodilation in ACLF, norepinephrine's vasoconstrictive efficacy may be more pronounced at the systemic level, thereby limiting its efficacy in reversing splanchnic vasodilation and associated HRS. It should also be noted that there are 2 key differences between the Arora et al trial and the other trials included in our analysis. In the Arora et al trial terlipressin was administered as continuous IV infusion as opposed to an IV bolus, and participants enrolled in the trial had lower SCr levels at baseline, both of which could have contributed to higher rates of HRS reversal [12]. These results align with a recent study which demonstrated that participants with lower SCr levels were more likely to experience better outcomes when treated with terlipressin compared to placebo [28]. Notably, participants from the Arora et al study observed better outcomes despite higher MELD scores at baseline [12]. Future studies of HRS treatments in the ACLF population could provide valuable data to further characterize the potential superiority of terlipressin over norepinephrine in this specific patient cohort.

It is important to note that there are key differences that may affect the choice of vasoconstrictor for HRS management. Norepinephrine must be given as a continuous IV infusion and typically requires a central venous line and, in most facilities, the transfer of the patient to an intensive care unit (ICU) [2]. In contrast, terlipressin can be given with a peripheral or central

line as a bolus or continuous infusion, allowing for use outside of the ICU setting [2,29,30]. Interestingly, most of the trials included in this analysis were conducted on medical floors (non-ICU), with the exception of 2 trials that treated all subjects in the ICU from the time HRS-1 diagnosis was suspected [16,17]. Most of the trials reported higher cost associated with the use of terlipressin but did not report costs of hospital admission, albumin, concomitant medications, and subsequent medical treatments after vasoconstrictor therapy [14–18].

The quality of evidence in our study was moderate based on exclusive inclusion of RCTs with objective endpoints such as HRS reversal and mortality. However, many trials were open-label [12,14–18], and did not report allocation concealment methods [15–17,19]. Although, blinding is unlikely to impact objective outcome measures such as HRS reversal and mortality. Of note, one of the trials in our analysis (Indrabi et al) was only published as an abstract, which could contribute to bias [19]. The majority of the studies included in our meta-analysis were conducted in India, which could limit the applicability of our results. A primary limitation in this analysis is the small number of studies and, as such, a small sample size. Additionally, each trial had slightly different diagnostic criteria for HRS-1, variation in dosing for terlipressin and norepinephrine, and throughout the time that these studies were conducted, the definition and nomenclature of HRS-1 changed. While HRS-1 diagnosis depends upon patients reaching a specific SCr threshold, the updated HRS-AKI diagnosis examines the change in SCr as the basis for diagnosis [1]. The updated HRS-AKI diagnostic criteria enable earlier treatment and therefore a better prognosis [30]. Because of their timing, all trials, except for Arora et al, did not use the updated definition of HRS-AKI for diagnosis. An additional limitation is that meta-analyses can be subject to misinterpretation due to heterogeneity related to considerable differences in the trials, which limit their comparability. We attempted to minimize this by including only RCTs of subjects with HRS-1 or HRS-AKI, which evaluated objective outcomes, and by using the random effects model to compensate for the heterogeneity. As more studies comparing terlipressin and norepinephrine are completed, a clearer understanding of the comparability of these 2 therapies will emerge.

In summary, our meta-analysis reveals a numerically higher rate of HRS reversal and lower rate of 1-month mortality for terlipressin compared to norepinephrine when results of these studies are considered in aggregate. Additionally, terlipressin was associated with an increased risk of AEs compared to norepinephrine, but discontinuation of therapy due to AEs was uncommon. Large head-to-head RCTs comparing the efficacy and safety of terlipressin and norepinephrine for the treatment of HRS-AKI as defined by recently revised guidelines would provide valuable insight to guide timely and effective therapy.

## Registration and protocol

A protocol was not prepared for this study. In addition, this study was not registered.

## Supporting information

**S1 Checklist. PRISMA 2020 for abstracts checklist.**
(DOCX)

**S2 Checklist. PRISMA 2020 checklist.**
(DOCX)

**S1 Table. Search strategies and selection criteria for studies.**
(DOCX)

**S2 Table. Review authors' judgments about each risk of bias item using Cochrane risk of bias assessment.**
(DOCX)

## Acknowledgments

Data extraction and statistical analysis were conducted by Andy Duydang, MS. Medical writing and editorial support, conducted in accordance with the International Committee of Medical Journal Editors and Good Publication Practice 3 guidelines, were provided by Kim Storvik, PhD, and Sanjna Patel, PharmD, of Red Nucleus, Yardley, PA, USA.

## Author Contributions

**Conceptualization:** Jody C. Olson, Ram M. Subramanian.

**Data curation:** Jody C. Olson, Ram M. Subramanian.

**Formal analysis:** Jody C. Olson, Ram M. Subramanian.

**Investigation:** Jody C. Olson, Ram M. Subramanian.

**Methodology:** Jody C. Olson, Ram M. Subramanian.

**Supervision:** Jody C. Olson, Ram M. Subramanian.

**Validation:** Jody C. Olson, Ram M. Subramanian.

**Visualization:** Jody C. Olson, Ram M. Subramanian.

**Writing – original draft:** Jody C. Olson, Ram M. Subramanian.

**Writing – review & editing:** Jody C. Olson, Ram M. Subramanian.

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
