## [Decision Letter · Decision Letter 0]

12 Sep 2023

PONE-D-23-19336Comparative Efficacy of Terlipressin and Norepinephrine for Treatment of Hepatorenal Syndrome-Acute Kidney Injury: A Systematic Review and Meta-analysisPLOS ONE

Dear Dr. Olson,

Thank you for submitting your manuscript to PLOS ONE. After careful consideration, we feel that it has merit but does not fully meet PLOS ONE’s publication criteria as it currently stands. Therefore, we invite you to submit a revised version of the manuscript that addresses the points raised during the review process.

We look forward to receiving your revised manuscript.

Kind regards,

Riccardo Nevola, MD, PhD

Academic Editor

PLOS ONE

“I have read the journal's policy and the authors of this manuscript have the following competing interests: [Both JCO and RMS have served as consultants for Mallinckrodt Pharmaceuticals related to terlipressin].”

Reviewers' comments:

Reviewer's Responses to Questions

**Comments to the Author**

1. Is the manuscript technically sound, and do the data support the conclusions?

Reviewer #1: Partly

Reviewer #2: Yes

Reviewer #3: Yes

2. Has the statistical analysis been performed appropriately and rigorously? 

Reviewer #1: I Don't Know

Reviewer #2: Yes

Reviewer #3: I Don't Know

3. Have the authors made all data underlying the findings in their manuscript fully available?

Reviewer #1: Yes

Reviewer #2: Yes

Reviewer #3: Yes

4. Is the manuscript presented in an intelligible fashion and written in standard English?

Reviewer #1: Yes

Reviewer #2: Yes

Reviewer #3: Yes

5. Review Comments to the Author

Reviewer #1: The authors have analysed the results of 7 RCTs comparing the use of terlipressin versus norepinephrine in combination with albumin in the treatment of HRS-AKI. They have introduced the need for such an analysis stating that there have been more studies after the last such analysis was published.

My comments:

1. Although the authors have been largely temperate in their recommendation of terlipressin over norepinephrine, it should be made clear that their conclusion is based on statistically insignificant differences. In a meta-analysis, that is an important point; technically, it should mean that the two interventions are equivalent and one cannot be promoted over the other. The interpretation should therefore be made balanced

2. It appears from Figures 2 and 3 that the study by Arora is an 'outlier' (if I may use that term). This study has a significant difference in methodology in that the terliprressin administration method was different and so was the endpoint. Does this skew the interpretation?

3. The study by Indrabi et al has too many missing datapoints and was available only as Abstract. Is it correct to include this in the meta-analysis for only a few parameters?

4. The authors have not mentioned the doses used of the two comparator drugs

5. Would the authors want to mention the comparative cost of the two drugs? I suspect that in most if not all countries, norepinephrine would be notably cheaper. I am not suggesting that the overall cost of treatment is less

6. (Minor): Ref 12 and 23 seem to be identical in authorship, title and journal.

Reviewer #2: Dear Editor,

The authors tried to focus on the comparison of terlipressin and norepinephrine in the setting of Hepatorenal syndrome- AKI. The approval of the use of terlipressin in this matter was recently approved by FDA and therefore this topic would be of greater interest in the US as well. Their main objective is to compare terlipressin with norepinephrine in terms of HRS reversal and short-term mortality.

I would suggest publishing this paper after some minor revisions as mentioned below.

Queries:

1. Sentence 103 says “Given the recent approval of terlipressin..”. However, even though it was recently approved in the US, it has been used in Europe for more than a few decades. I would suggest adjusting that sentence.

2. Sentence 113 says “They also found that terlipressin may reduce short-term mortality compared to placebo”. Could you please add a small info regarding norepinephrine’s impact on short-term mortality here?

3. It kept saying that “except for Indrabi et al [19], which did not disclose the treatment duration in its abstract”. It looks like only the abstract of this study was reachable! And the risk of bias came high apparently! It would have been a good choice to remove this study from the analysis!

4. I don’t know what it is the main reason, but per the figures, Arora’s study provided significantly favorable results for the terlipression arm! Per the demographic data, Arora’s patients’ MELD scores seemed to be higher and creatinine levels lower! One study with such a significant skew on behalf of terlipressin could be a game changer as happened in this meta-analysis! Any comments about this? I don’t know if continuous vs bolus administration of terlipression can make such a huge difference.

Reviewer #3: I have two main criticisms of this meta-analysis.

1- Six of the 7 studies included in the meta-analysis were from India. Doesn't this question the universality of the results?

2- In only one of the 7 studies included in the meta-analysis (Arora et al), terlipressin was used as an infusion. With this method, you can achieve continuous effectiveness by using a much smaller amount of terlipressin. Looking at the tables, in the Arora et al. study, HRS reversal was 3.35 times better and 1-month mortality was 3.74 times better in continious terlipressin infussion when comperad noradrenalin. So, the study that determined that terlipressin was better in this meta-analysis was the study of Arora et al. Other studies don't support this much, leaving it questionable. It should be emphasized that it may be more appropriate to use terlipressin as an infusion.

6. PLOS authors have the option to publish the peer review history of their article (what does this mean?). If published, this will include your full peer review and any attached files.

Reviewer #1: No

Reviewer #2: **Yes: **Akif Altinbas

Reviewer #3: **Yes: **Huseyin Alkim

Best regards

---

## [Author Response · Author response to Decision Letter 0]

6 Nov 2023

Reviewer 1

1. “Although the authors have been largely temperate in their recommendation of terlipressin over norepinephrine, it should be made clear that their conclusion is based on statistically insignificant differences. In a meta-analysis, that is an important point; technically, it should mean that the two interventions are equivalent and one cannot be promoted over the other. The interpretation should therefore be made balanced.”

a. Response: Thank you for your suggestion and we agree. We have updated the manuscript accordingly. 

b. Revision (PG20/P1/L322-324): “In our assessment of the new studies being added to this meta-analysis, we chose to remove the study by Nayyar et al because the results fell significantly outside of the normal distribution in favor of terlipressin. Even without this study, data from our meta-analysis numerically favor terlipressin use, although it should be noted that the difference between terlipressin and norepinephrine for both HRS reversal and mortality are not statistically significant. Our conclusion is similar to the conclusion reported by Zheng et al, who also reported that the difference between norepinephrine and terlipressin was not statistically significant but used a graphical representation of SUCRA probability to demonstrate that terlipressin was considered the most effective for HRS reversal.”

Revision (PG21/P2/L364-365, PG22/P1/L366) “While the data presented here numerically favor the use of terlipressin over norepinephrine, it is important to note that there are key differences that may affect the choice of vasoconstrictor for HRS management.”

2. “It appears from Figures 2 and 3 that the study by Arora is an “outlier” (if I may use that term). This study has a significant difference in methodology in that the terlipressin administration method was different and so was the endpoint. Does this skew the interpretation?” 

a. Response: Thank you for highlighting the differences in the Arora et al study compared to other studies included in our analysis. We agree that terlipressin administration with intravenous infusion may result in better outcomes for participants with hepatorenal syndrome (HRS) as seen in the Cavallin et al study. [1] In addition, the difference in the study endpoint may have also impacted our data. However, we compensated for this heterogeneity via our statistical analysis of the data. As stated in lines 164-166, we used a random effects model with Knapp-Hartung adjustment to evaluate the pooled effect sizes between the studies. The random effects model takes into consideration the differences in the target population, intensity and length of treatment, or other differences. As such, we feel that these differences could not have skewed the interpretation significantly. 

The Arora et al study may look like an outlier, but it is important to examine the results in the context of all available data. For example, when the Nayyar et al study was included in our analysis, it significantly skewed our results calling into question its reliability as shown in Fig 1, attached in the "Response to Reviewers" document included in the submission. [2]

Fig 1. Meta-analysis comparing HRS reversal of norepinephrine and terlipressin. Forest plot matrix that compares reported HRS reversal data of all included trials using OR and heterogeneity calculated with a random effects model by pairwise meta-analysis. CI = confidence interval, OR = odds ratio, SE = standard error.

Fig 1 demonstrates how the Nayyar et al study skewed the results since it falls significantly outside the normal distribution in the forest plot matrix. Based on the P value of the Heterogeneity (I2), the effect size is small, indicating limited practical applications; hence, it was excluded from our analysis. In comparison to the Nayyar et al study, the Arora et al study is not an outlier in this context. 

In addition, due to the limited studies that met our inclusion criteria, our sample size was small. The Arora et al study was one of the most recent and larger studies among others in our analysis. We felt it was important to include all eligible studies, especially since previous meta-analyses did not include the Arora et al trial. Addition of newer trials could provide further insight into the comparative efficacy of terlipressin and norepinephrine.

b. References: 

1. Cavallin M, Piano S, Romano A, Fasolato S, Frigo AC, Benetti G, et al. Terlipressin given by continuous intravenous infusion versus intravenous boluses in the treatment of hepatorenal syndrome: a randomized controlled study. Hepatology. 2016;63(3):983-992. doi: 10.1002/hep.28396.

2. Nayyar S, Kaur R, Mohan G, Chandey M. A prospective study to compare the efficacy of noradrenaline verses terlipressin in hepatorenal syndrome in patients with advanced cirrhosis. Int J Adv Med. 2021;8(9):1312-1318. doi: 10.18203/2349-3933.ijam20213215.

3. “The study by Indrabi et al has too many missing datapoints and was available only as Abstract. Is it correct to include this in the meta-analysis for only a few parameters?”

a. Response: Thank you for your critique. We agree that the Indrabi et al study has missing data and methodology since only the abstract was available. However, we felt it was important to include all studies from previous meta-analyses, such as Facciorusso et al and Wang et al, if we were able to extract the data. [1,2] Our statistician was able to extract the data for the HRS reversal endpoint in our meta-analysis. However, we were unable to extract mortality data despite contacting the authors directly. As such, the Indrabi et al study is not included in the short-term mortality endpoint. 

In addition, there are very limited studies that conducted a comparative evaluation of terlipressin and norepinephrine in patients with HRS. Because our sample size was small, we felt it was important to include all eligible studies for analysis, especially if they were discussed in prior meta-analyses. 

b. References: 

1. Facciorusso A, Chandar AK, Murad MH, et al. Comparative efficacy of pharmacological strategies for management of type 1 hepatorenal syndrome: a systematic review and network meta-analysis. Lancet Gastroenterol Hepatol. 2017;2(2):94-102. doi: 10.1016/S2468-1253(16)30157-1.

2. Wang H, Liu A, Bo W, Feng X, Hu Y. Terlipressin in the treatment of hepatorenal syndrome: a systematic review and meta-analysis. Medicine (Baltimore). 2018;97(16):e0431. doi: 10.1097/MD.0000000000010431.

4. “The authors have not mentioned the doses used of the two comparator drugs”

a. Response: Thank you for your feedback. We acknowledge that the dosing for both terlipressin and norepinephrine are not discussed in detail throughout the manuscript; however, the specific doses for both terlipressin and norepinephrine are included in Table 1 of the manuscript. Despite slight variations among the studies, dosing for both drugs were comparable to one another and other studies that have evaluated patients with HRS. However, we do realize that the variation in dosing could contribute to heterogeneity and is a limitation of the analysis. As such, we have revised the manuscript accordingly.

b. Revision (PG22/P2/L381-385): “A primary limitation in this analysis is the small number of studies and, as such, a small sample size. Additionally, each trial had slightly different diagnostic criteria for HRS-1, variation in dosing for terlipressin and norepinephrine, and throughout the time that these studies were conducted, the definition and nomenclature of HRS-1 changed.”

5. “Would the authors want to mention the comparative cost of the two drugs? I suspect that in most if not all countries, norepinephrine would be notably cheaper. I am not suggesting that the overall cost of treatment is less” 

a. Response: Thank you for highlighting this important variable. However, we opted not to discuss the comparative cost of the two drugs because the objective of our analysis was to evaluate the safety and efficacy of terlipressin in comparison to norepinephrine among patients with HRS. While cost will be an important factor for hospitals, it is not part of this scientific evaluation.

6. “(Minor) Ref 12 and 23 seem to be identical in authorship, title and journal.”

a. Response: Thank you for highlighting this. However, please note references 12 and 23 cite the Arora et al study and the corresponding Arora et al supplement, which is why they appear identical with the exception of ‘(suppl)’ after the volume and issue. We have revised the references according to Vancouver style.

b. Revision (PG26/Reference 12, PG27/Reference 23): 

1. Arora V, Maiwall R, Rajan V, Jindal A, Shasthry SM, Kumar G, Jain P, Sarin SK Terlipressin is superior to noradrenaline in the management of acute kidney injury in acute on chronic liver failure. Hepatology. 2020;71(2):600-610. 

2. Arora V, Maiwall R, Rajan V, Jindal A, Shasthry SM, Kumar G, Jain P, Sarin SK. Terlipressin is superior to noradrenaline in the management of acute kidney injury in acute on chronic liver failure. Hepatology. 2020;71(2)(suppl):1-5. 

Reviewer 2

1. “Sentence 103 says ‘Given the recent approval of terlipressin..’ However, even though it was recently approved in the US, it has been used in Europe for more than a few decades. I would suggest adjusting that sentence.” 

a. Response: Thank you for this suggestion. We agree and have revised the sentence to specify the recent approval occurred in the US in lines 90-93. 

b. Revision (PG5/P1/L90-93): “Given the recent approval of terlipressin in the US and the historical use of norepinephrine in patients with HRS-AKI, there is a need among US physicians to evaluate the comparative efficacy of these treatment options commonly used as first-line therapy”

2. “Sentence 113 says ‘They also found that terlipressin may reduce short-term mortality compared to placebo’. Could you please add a small info regarding norepinephrine’s impact on short-term mortality here?”

a. Response: Thank you for highlighting this, we have added an additional sentence to discuss the short-term mortality data for norepinephrine in lines 99-101. 

b. Revision (PG5/P2/L99-101): “They also found that terlipressin may reduce short-term mortality compared to placebo; but neither terlipressin nor norepinephrine was associated with a significant mortality benefit over any comparator. [9]”

3. “It kept saying that ‘except for Indrabi et al, [19] which did not disclose the treatment duration in its abstract.’ It looks like only the abstract of this study was reachable! And the risk of bias came high apparently! It would have been a good choice to remove this study from the analysis!”

a. Response: Thank you for your critique. We agree that the Indrabi et al study has missing information, which could have contributed to bias. However, we felt it was important to include all studies from previous meta-analyses, such as Facciorusso et al and Wang et al, if we were able to extract the data. [1,2] Our statistician was able to extract the data for the HRS reversal endpoint in our meta-analysis. However, we were unable to extract mortality data despite contacting the authors directly. As such, the Indrabi et al study is not included in the short-term mortality endpoint. 

In addition, due to the limited studies that met our inclusion criteria, our sample size was small. Therefore, we felt it was important to include all eligible studies for analysis, especially if they were included in prior meta-analyses to help evaluate the comparative efficacy and safety of terlipressin and norepinephrine. 

b. References: 

1. Facciorusso A, Chandar AK, Murad MH, et al. Comparative efficacy of pharmacological strategies for management of type 1 hepatorenal syndrome: a systematic review and network meta-analysis. Lancet Gastroenterol Hepatol. 2017;2(2):94-102. doi: 10.1016/S2468-1253(16)30157-1.

2. Wang H, Liu A, Bo W, Feng X, Hu Y. Terlipressin in the treatment of hepatorenal syndrome: a systematic review and meta-analysis. Medicine (Baltimore). 2018;97(16):e0431. doi: 10.1097/MD.0000000000010431.

4. “I don’t know what it is the main reason, but per the figures, Arora’s study provided significantly favorable results for the terlipressin arm! Per the demographic data, Arora’s patients’ MELD scores seemed to be higher and creatinine levels lower! One study with such a significant skew on behalf of terlipressin could be a game changer as happened in this meta-analysis! Any comments about this? I don’t know if continuous vs bolus administration of terlipressin can make such a huge difference.” 

a. Response: Thank you for providing your feedback. We agree that terlipressin administration and lower baseline serum creatinine (SCr) levels (as shown in Table 2 of the manuscript) could have skewed the results and contributed to better outcomes. But this likely would have been true for both treatments. We have revised the manuscript to acknowledge this variable (lines 353-357). 

We also agree that participants in the Arora et al study had higher model for end-stage liver disease (MELD) scores compared to other participants as discussed in Table 2 of the manuscript. However, we believe that higher MELD scores would probably not have skewed the results in the Arora et al study toward higher HRS reversal, but rather these participants experienced better outcomes despite high MELD scores. Pooled data from 3 trials OT-0401, REVERSE, and the CONFIRM, which evaluated the safety and efficacy of terlipressin in comparison to placebo support the assertion that patients with higher MELD scores do not experience increased HRS reversal rates, as shown in Table 1, attached in the "Response to Reviewers" document included in the submission. [1]

Table 1. Rate of HRS Reversal by MELD Score in the Pooled Population (Intent-to-Treat) [1] 

 Terlipressin Placebo

Baseline MELD scorea N n (%) N n (%)

Low MELD score 145 63 (43.4) 101 20 (19.8)

High MELD score 167 40 (24.0) 120 17 (14.2)

a Please note, the median MELD determines high/low MELD score. A low MELD score is <34. A high MELD score is ≥34. 

b. Revision (PG21/P1/L353-357): “It should also be noted that there are 2 key differences between the Arora et al trial and the other trials included in our analysis. In the Arora et al trial terlipressin was administered as continuous IV infusion as opposed to an IV bolus, and participants enrolled in the trial had lower SCr levels at baseline, both of which could have contributed to higher rates of HRS reversal. [12]” 

c. Reference: 

1) FDA Cardiovascular and Renal Drugs Advisory Committee. Mallinckrodt Pharmaceuticals Terlipressin Advisory Committee Briefing Document. NDA #022231. July 2020. Accessed October 6, 2023. https://public4.pagefreezer.com/browse/FDA/05-05-2022T12:59/https:/www.fda.gov/media/139965/download

Reviewer 3

1. “Six of the 7 studies included in the meta-analysis were from India. Doesn’t this question the universality of the results?”

a. Response: Thank you for bringing up this critique. We agree that the location where the studies were conducted limit the applicability of our data. However, there are a very small number of studies that evaluate the safety and efficacy of norepinephrine and terlipressin. Our inclusion criteria did not restrict studies according to location and the thorough literature search from the past 16 years did not yield other eligible studies. This is one of the limitations of our meta-analysis, and we have updated the manuscript to reflect this in lines 380-381. 

b. Revision (PG22/P2/L380-381): “The majority of the studies included in our meta-analysis were conducted in India, which could limit the applicability of our results.”

2. “In only one of the 7 studies included in the meta-analysis (Arora et al), terlipressin was used as an infusion. With this method, you can achieve continuous effectiveness by using a much smaller amount of terlipressin. Looking at the tables, in the Arora et al. study, HRS reversal was 3.35 times better and the 1-month mortality was 3.74 times better in continuous terlipressin infusion when compared noradrenalin. So, the study that determined that terlipressin was better in this meta-analysis was the study of Arora et al. Other studies don’t support this much, leaving it questionable. It should be emphasized that it may be more appropriate to use terlipressin as an infusion.” 

a. Response: Thank you for making this important point about the Arora manuscript. We agree that both the results and the study size make it a substantial contributor to the analysis. While continuous infusion is an important differentiator for this study, there are other factors discussed in the paper as well, which could have contributed to the rates of HRS reversal and mortality. We do think highlighting that it was continuous infusion is important, but we do not feel that there is sufficient evidence to draw a conclusion about it being a more appropriate route of administration, and this would be considered off-label use. [1] As such, we have updated the copy as follows:

b. Revision (PG21/P1/L353-357): “It should also be noted that there are 2 key differences between the Arora et al trial and the other trials included in our analysis. In the Arora et al trial terlipressin was administered as continuous IV infusion as opposed to an IV bolus, and participants enrolled in the trial had lower SCr levels at baseline, both of which could have contributed to higher rates of HRS reversal. [12]”

c. Reference: 

1) TERLIVAZ® (terlipressin). Prescribing Information. Bridgewater, NJ: Mallinckrodt Hospital Products Inc.

---

## [Decision Letter · Decision Letter 1]

24 Nov 2023

PONE-D-23-19336R1Comparative efficacy of terlipressin and norepinephrine for treatment of hepatorenal syndrome-acute kidney injury: a systematic review and meta-analysisPLOS ONE

Dear Dr. Olson,

Thank you for submitting your manuscript to PLOS ONE. After careful consideration, we feel that it has merit but does not fully meet PLOS ONE’s publication criteria as it currently stands. Therefore, we invite you to submit a revised version of the manuscript that addresses the points raised during the review process.

We look forward to receiving your revised manuscript.

Kind regards,

Riccardo Nevola, MD, PhD

Academic Editor

PLOS ONE

Journal Requirements:

Reviewers' comments:

Reviewer's Responses to Questions

**Comments to the Author**

1. If the authors have adequately addressed your comments raised in a previous round of review and you feel that this manuscript is now acceptable for publication, you may indicate that here to bypass the “Comments to the Author” section, enter your conflict of interest statement in the “Confidential to Editor” section, and submit your "Accept" recommendation.

Reviewer #1: (No Response)

Reviewer #2: All comments have been addressed

Reviewer #3: All comments have been addressed

2. Is the manuscript technically sound, and do the data support the conclusions?

Reviewer #1: Partly

Reviewer #2: Yes

Reviewer #3: Yes

3. Has the statistical analysis been performed appropriately and rigorously? 

Reviewer #1: I Don't Know

Reviewer #2: Yes

Reviewer #3: I Don't Know

4. Have the authors made all data underlying the findings in their manuscript fully available?

Reviewer #1: Yes

Reviewer #2: Yes

Reviewer #3: Yes

5. Is the manuscript presented in an intelligible fashion and written in standard English?

Reviewer #1: Yes

Reviewer #2: Yes

Reviewer #3: Yes

6. Review Comments to the Author

Reviewer #1: Although the authors have tried to be temperate in their text, there are still several places where a 'superiority' of terlipressin has crept in in the wordings. The authors clarify that in all these places, there is no statistical difference. In that case, the authors should clearly state only such a message. In the plots provided (Figs 2 and 3), the study by Arora et al is the only one that is unequivocally in favour of terlipressin and this study has the maximum weightage. There are notable differences in the methodology in this study as compared to the other studies

Reviewer #2: Dear Editor,

In my opinion, this article with its current form is ready for publication. I would suggest publishing it without further revision.

I appreciate for the authors' time to make adjustments based on our suggestions.

Best regards

Reviewer #3: Appropriate responses were given to all my criticisms. I have no additional criticism. It can be published in this form.

7. PLOS authors have the option to publish the peer review history of their article (what does this mean?). If published, this will include your full peer review and any attached files.

Reviewer #1: No

Reviewer #2: **Yes: **Akif Altinbas

Reviewer #3: **Yes: **Huseyin ALKIM

---

## [Author Response · Author response to Decision Letter 1]

8 Dec 2023

Response Letter 

Reviewer 1

1. “Although the authors have tried to be temperate in their text, there are still several places where a 'superiority' of terlipressin has crept in in the wordings. The authors clarify that in all these places, there is no statistical difference. In that case, the authors should clearly state only such a message. In the plots provided (Figs 2 and 3), the study by Arora et al is the only one that is unequivocally in favour of terlipressin and this study has the maximum weightage. There are notable differences in the methodology in this study as compared to the other studies.”

a. Response: Thank you for your feedback. We have updated the manuscript accordingly. 

b. Revision (PG2/P1/L41-43, PG3/P1/L44): “This meta-analysis showed numerically higher rates of HRS reversal (OR 1.33, 95% confidence interval [CI] [0.80-2.22]; P=0.22) and short-term survival (OR 1.50, 95% CI [0.64-3.53]; P=0.26) with terlipressin, though these results did not reach statistical significance.”

Revision (PG3/P1/L48-49): “Limitations of this analysis included small sample size and study differences in HRS-AKI diagnostic criteria.”

Revision (PG15/P1/L226-227): “However, these results were not statistically significant (OR 1.33; 95% CI [0.80-2.22]; P=0.22) (Fig 2).”

Revision (PG16/P1/L244): “However, the result was not statistically significant (OR 1.50, 95% CI [0.64-3.53]; P=0.26; Fig 3).”

Revision (PG18/P2/L297-300): “Our data support this recommendation, with terlipressin having higher rates of both HRS reversal and short-term survival, although the observed ORs for these endpoints did not achieve statistical significance, suggesting norepinephrine is an appropriate option as noted in the guidelines.”

Revision (PG19/P3/L322, PG20/P1/L323-325): “Even without this study, our conclusion is similar to that reported by Zheng et al, who also reported that the difference between norepinephrine and terlipressin was not statistically significant but used a graphical representation of SUCRA probability to demonstrate that terlipressin was considered the most effective for HRS reversal.”

Revision (PG21/P2/L360-361): “It is important to note that there are key differences that may affect the choice of vasoconstrictor for HRS management.”

Revision (PG23/P2/L391-393): “In summary, our meta-analysis reveals a numerically higher rate of HRS reversal and lower rate of 1-month mortality for terlipressin compared to norepinephrine when results of these studies are considered in aggregate.”

---

## [Editor Report · Decision Letter 2]

18 Dec 2023

Comparative efficacy of terlipressin and norepinephrine for treatment of hepatorenal syndrome-acute kidney injury: A systematic review and meta-analysis

PONE-D-23-19336R2

Dear Dr. Jody Olson,

We’re pleased to inform you that your manuscript has been judged scientifically suitable for publication and will be formally accepted for publication once it meets all outstanding technical requirements.

Kind regards,

Riccardo Nevola, MD, PhD

Academic Editor

PLOS ONE

---

## [Editor Report · Acceptance letter]

19 Jan 2024

PONE-D-23-19336R2 

PLOS ONE

Dear Dr. Olson, 

I'm pleased to inform you that your manuscript has been deemed suitable for publication in PLOS ONE. Congratulations! Your manuscript is now being handed over to our production team.

Kind regards, 

on behalf of

Dr. Riccardo Nevola 

Academic Editor

PLOS ONE